# Radiomic Feature Reduction Approach to Predict Breast Cancer by Contrast-Enhanced Spectral Mammography Images

**DOI:** 10.3390/diagnostics11040684

**Published:** 2021-04-10

**Authors:** Raffaella Massafra, Samantha Bove, Vito Lorusso, Albino Biafora, Maria Colomba Comes, Vittorio Didonna, Sergio Diotaiuti, Annarita Fanizzi, Annalisa Nardone, Angelo Nolasco, Cosmo Maurizio Ressa, Pasquale Tamborra, Antonella Terenzio, Daniele La Forgia

**Affiliations:** 1Struttura Semplice Dipartimentale di Fisica Sanitaria, I.R.C.C.S. Istituto Tumori “Giovanni Paolo II”, Viale Orazio Flacco 65, 70124 Bari, Italy; r.massafra@oncologico.bari.it (R.M.); mariac.comes@libero.it (M.C.C.); v.didonna@oncologico.bari.it (V.D.); p.tamborra@oncologico.bari.it (P.T.); 2Dipartimento di Matematica, Università degli Studi di Bari, 70121 Bari, Italy; s.bove9@studenti.uniba.it; 3Unità Operativa Complessa di Oncologia Medica, I.R.C.C.S. Istituto Tumori “Giovanni Paolo II”, Viale Orazio Flacco 65, 70124 Bari, Italy; vitolorusso@me.com (V.L.); angelonolasco@hotmail.com (A.N.); 4Dipartimento di Economia e Finanza, Università degli Studi di Bari, 70124 Bari, Italy; a.biafora@studenti.uniba.it; 5Struttura Semplice Dipartimentale di Chirurgia, I.R.C.C.S. Istituto Tumori “Giovanni Paolo II”, Viale Orazio Flacco 65, 70124 Bari, Italy; sergiodiotiaiuti@gmail.com; 6Unita Opertiva Complessa di Radioterapia, IRCCS Istituto Tumori ”Giovanni Paolo II”, 70124 Bari, Italy; a.nardone@oncologico.bari.it; 7Unità Operativa Complessa di Chirurgica Plastica e Ricostruttiva, I.R.C.C.S. Istituto Tumori “Giovanni Paolo II”, Viale Orazio Flacco 65, 70124 Bari, Italy; m.ressa@oncologico.bari.it; 8Unità di Oncologia Medica, Università Campus Bio-Medico, 00128 Roma, Italy; a.terenzio@unicampus.it; 9Struttura Semplice Dipartimentale di Radiologia Senologica, I.R.C.C.S. Istituto Tumori “Giovanni Paolo II”, Viale Orazio Flacco 65, 70124 Bari, Italy; d.laforgia@oncologico.bari.it

**Keywords:** breast cancer, contrast-enhanced spectral mammography (CESM), principal component analysis (PCA), computer-automated diagnosis (CADx), feature extraction, feature reduction

## Abstract

Contrast-enhanced spectral mammography (CESM) is an advanced instrument for breast care that is still operator dependent. The aim of this paper is the proposal of an automated system able to discriminate benign and malignant breast lesions based on radiomic analysis. We selected a set of 58 regions of interest (ROIs) extracted from 53 patients referred to Istituto Tumori “Giovanni Paolo II” of Bari (Italy) for the breast cancer screening phase between March 2017 and June 2018. We extracted 464 features of different kinds, such as points and corners of interest, textural and statistical features from both the original ROIs and the ones obtained by a Haar decomposition and a gradient image implementation. The features data had a large dimension that can affect the process and accuracy of cancer classification. Therefore, a classification scheme for dimension reduction was needed. Specifically, a principal component analysis (PCA) dimension reduction technique that includes the calculation of variance proportion for eigenvector selection was used. For the classification method, we trained three different classifiers, that is a random forest, a naïve Bayes and a logistic regression, on each sub-set of principal components (PC) selected by a sequential forward algorithm. Moreover, we focused on the starting features that contributed most to the calculation of the related PCs, which returned the best classification models. The method obtained with the aid of the random forest classifier resulted in the best prediction of benign/malignant ROIs with median values for sensitivity and specificity of 88.37% and 100%, respectively, by using only three PCs. The features that had shown the greatest contribution to the definition of the same were almost all extracted from the LE images. Our system could represent a valid support tool for radiologists for interpreting CESM images.

## 1. Introduction

Reducing the breast cancer mortality rate in the population of women worldwide is greatly impacted by early diagnosis using mammography [1,2].

In recent years, new mammography techniques such as contrast-enhanced spectral mammography (CESM) and tomosynthesis have contributed to further improve the performance of mammography, even in less “readable” types of breasts due to their high density [3,4].

Specifically, the CESM represents a unique and very particular methodology in the clinical and scientific field due to the considerable amount of information derived from morphological images and contrast dynamics obtained simultaneously on the same breast and the same region of interest.

While CESM has the same type of indications as DCE-MRI due to the radiation dose and non-uniform availability of the territory, CESM is currently used according to the guidelines as an alternative to DCE-MRI in cases where this is not feasible (for absolute contraindications or related) or not available [5].

In MRI the performance of the human reader can be reduced by the presence of marked background parenchymal enhancement (BPE), defined as the normal background impregnation of the breast but with very different intensity and distribution in relation to age, hormonal phase, menopausal state and any therapies in progress [6]. In these circumstances, diagnostic aids to reporting and fusion-imaging systems in different positions can support the diagnosis [7,8,9]. These aspects are also expected in CESM.

Nowadays, clinicians are aided in the interpretation of these images by automated support systems, known as computer-aided detection/diagnosis (CAD), which is software used in clinical medicine to suggest diagnoses and treatments based on the clinical data and algorithms for their interpretation. Although various models of CAD systems for breast diseases using mammographic images have been developed in the past years [10,11,12,13,14,15], the literature is limited with regard to CAD systems for breast lesions using CESM images. Most of the recent literature includes comparative studies on the diagnostic performance of human readers on CESM images with respect to those of mammographic and MR images [3,16,17,18]. However, interest in this new instrumentation is strong and some support tools have recently been proposed for characterizing breast lesions [19,20,21,22,23] and predicting the response to neoadjuvant therapy [24,25].

For this reason, we have developed an accurate computer-aided diagnosis tool to classify breast cancer lesions based on radiomic analysis of CESM images. In our previous work, we proposed a preliminary radiomics analysis aimed to explore the usefulness of quantitative information extracted from CESM images, to understand the behavior of each different set of well-known textural features automatically extracted from CESM images, and to compare them with each other [21,23]. In these works, an important role was played by the feature selection processes used to describe and characterize the regions of interest (ROIs) identified by our expert radiologists. In order to reduce the dataset dimensionality, starting from the initial feature set, a sub-set of these features, which were characterized by their high discriminating power, was selected by filtering techniques (i.e., statistical tests) for more manageable data processing [26]. Then, we selected the most important features by developing different approaches to feature selection, such as embedded and wrapper methods.

The feature selection techniques can influence the process and accuracy of cancer classification. Indeed, although the subset of features identified was the one with the greatest discriminating power with respect to the various evaluation criteria considered, the feature selection techniques involve a natural and inevitable loss of information due to the exclusion of features from the original dataset. For this reason, before proceeding with the exploration of new features useful for increasing the accuracy of the classification performance, we considered it appropriate to evaluate a different dimensionality reduction technique that does not produce information losses but only a reduction of the noise inherent in the dataset.

Therefore, we evaluated a classification scheme that includes dimension reduction. Specifically, firstly, a principal component analysis (PCA) dimension reduction method that includes the calculation of the proportion of variance for eigenvector selection was used. Following, the ROIs classification was performed by training different binary classifiers on a subset identified by sequential supervised learning procedures.

## 2. Materials and Methods

### 2.1. Materials

#### 2.1.1. CESM Examination

The CESM is based on a dual-energy exposure after a single injection of an iodinated contrast medium (CM), which produces three images: a low-energy (LE), a high-energy (HE) and a third image, defined as a recombined (RC) image that is obtained thanks to the digital subtraction of the LE image from the HE one.

Finally, two images are displayed by the radiologist on the reporting monitor for diagnostic purposes: the LE, which can be superimposed on a standard 2D mammography, which allows morphological surveys, and the RC, which provides information on the tumor neoangiogenesis of the breast.

An example of CESM images is shown in Figure 1, which includes a LE image (a), a HE image (b) and their RC image (c).

For all CESM exams, a modified Full-Field Digital Mammography (FFDM) system derived from a standard Senographe Essential (GE Healthcare) was used. First, the breast with no pathology was imaged, then the breast with the suspected lesion. Both craniocaudal (CC) and mediolateral oblique (MLO) views were collected. All of the images obtained were in DICOM format and were evaluated by a dedicated radiologist with more than 10 years of experience in reading mammography and breast MR images and trained in reading contrast-enhanced images.

#### 2.1.2. Experimental Dataset

The study was pre-approved by the Scientific Board of the Istituto Tumori “Giovanni Paolo II” of Bari, Italy. As this is a retrospective study, the anonymized images of patients who had given consent to the use of data for scientific purposes, as required by our Regulations, were acquired.

Once we had selected images from 53 patients aged between 37 and 76 years (with a mean of 52.2 ± 10.1 years), showing a positive result according to CESM, a radiologist from our Institute dedicated to senologic diagnostics manually identified the ROIs with a box on the image by using the reporting tool. Some patients had more than one, so a total of 58 ROIs were identified according to the BIRADS classification [27]: lesions belonging to BIRADS 2 and 3 classes were considered as benign, while lesions belonging to BIRADS 4 and 5 classes were labeled as malignant. Then, the histological diagnosis based on bioptic sampling established that 15 ROIs contained benign lesions while 43 ROIs included malignant ones.

### 2.2. Methods

As summarized in Figure 2, after the radiologist manually segmented a region of interest (ROI) from each LE and RC image, a large feature set consisting of five different kinds was extracted, such as points and corners of interest, textural and statistical features from both the original ROIs and also the ones obtained by a Haar decomposition and a gradient image implementation. Subsequently, principal component analysis (PCA) dimension reduction method that includes the calculation of variance proportion for eigenvector selection was used from each of five features set. Finally, different binary classifiers were trained to discriminate benign and malignant ROIs by developing a sequential forward feature selection algorithm that selected feature sub-sets first individually, and then simultaneously. MATLAB R2017a (MathWorks, Inc., Natick, MA, USA) software was used for all analyses.

#### 2.2.1. Feature Extraction

Starting from each ROI extracted from both original LE and RC images, five feature sets were automatically extracted to make the lesion classification more objective and operator independent. We then started to mathematically define a digital image [28].

An image can be defined as a two-dimensional function mapping the spatial coordinates x and y into a value f(x,y) representing the intensity of gray level of the image at that point. When x, y, and f(x,y) are all finite and discrete quantities, we call the image a digital image. A digital image is composed of a finite number of elements called pixels, each one having a particular position and gray intensity value. The section of the real plane spanned by the coordinates of the image is called the spatial domain. A very common representation for a digital image is a two-dimensional array represented as an M×N numerical matrix.

Therefore, given a digital image represented by a matrix, it was possible to extract statistical and textural features and, moreover, to identify points, edges and corners of interest. As shown in Figure 3, in this work we used five different extracted feature subsets, which we named as follows: STAT set, COUNT set, GRAD set, HAAR set and GLCM set [21,23].

The STAT set consists of 22 statistical features that describe the distribution of the ROI gray levels, measuring the likelihood of observing a gray value in random positions in the image, without taking into account the spatial information [29]. From each LE and RC original ROI, we extracted the following features: mean, standard deviation and their ratio, variance, skewness, entropy, relative smoothness, kurtosis, minimum and maximum values of gray-level and their difference.

The COUNT set contains a total of 10 features that describe the points, edges and corners of interest. In this work, we used five known learning algorithms:Scale invariant feature transform (SIFT) algorithm [30,31], which detects and describes image local featuresMinimum eigenvalue algorithm, which underlies the Shi-Tomasi corner detection algorithm [32] for identifying the corners of an objectFeatures from the accelerate segment test (FAST) algorithm [33,34], which is another corner detection methodBinary robust invariant scalable key-points (BRISK) method [35], which combines the SIFT and the FAST algorithms to feature detection, descriptor composition and key-points matchingMaximally stable external regions (MSER) algorithm [36], which is a method of blob detection in images whose aim consists of finding correspondence between image elements from two images with different viewpoints.

The GRAD set is formed by 24 features and contains some previously defined statistical features, that is, mean, variance, skewness, entropy, relative smoothness and kurtosis, extracted from the gradient’s magnitude and direction of each LE and RC original ROI. The gradient of an image is represented as a two-component vector (x- and y-derivative) defined at each pixel [28]. These can be computed by the convolution with a kernel, such as the Sobel or Prewitt operator, since the image is a discrete function for which the derivatives are not defined. For each vector, the magnitude *Gmag* shows how quickly the intensity of each pixel is changing in the neighborhood of the pixel (x, y) in the direction of the gradient, while the direction *Gdir* represents the orientation of greatest intensity change in the neighborhood of the pixel (x, y). The gradient can be approximated by convolving a kernel, in this work a Sobel kernel, with the original image [28]. The importance of calculating the gradient image lies in the two pieces of information it provides: the magnitude, which is a measure of how quickly the image is changing and the direction, which illustrates the direction in which the image is changing most rapidly.

The HAAR set contains 96 features, the same statistical features previously computed in the GRAD set, but this time extracted from each sub-ROI obtained by decomposing each LE and RC original ROI thanks to the Haar wavelet transform [28,37]. This technique obtains multi-resolution representations of images, which are very effective for analyzing the information content of images due to the dependence of the texture on the scale at which an image is analyzed. Particularly, once Haar two-dimensional scaling and wavelets functions are computed, these can be used as filters in order to decompose the image into four bands. First the image is low-pass filtered and downscaled in order to obtain the low low (LL1) band, then it is high-pass filtered in the three different directions in order to obtain the three types of detail images: horizontal (HL1), vertical (LH1) and diagonal (HH1). The operations can then be repeated on the LL1 band using the identical filters, as shown in Figure 4.

Finally, the GLCM set is formed by a total of 312 features, which represent the spatial relationship that on average links the gray levels of the image to each other. Texture analysis returns textural variables, which are any geometric and repetitive arrangement of gray levels. The textural information of an image can be described by second-order variables computed on the N × N gray-level co-occurrence matrix (GLCM) (N represents the number of intensity values in the gray-scale image, which is usually reduced from 256 to 8 at the beginning of the algorithm to reduce the computational cost) [38,39].

Given a gray-level image f, the first step in building the matrix consists in defining a specific spatial relationship between pixels. This relationship, known as offset, is the distance D between a pixel of interest and its neighbors with respect to a specific direction, identified by an angle θ ∈ {0°,45°,90°,135°}. Thus, the (i,j) element in GLCM matrix represents the number of times the combination of level i and j occurs in two pixels in the image, which satisfy the relationship given by the offset [28].

Thus, for each sub-ROI previously decomposed by the Haar transform only at first level, the co-occurence matrices are extracted in the four possible directions by choosing the parameter D equal to 2 in order to evaluate the relationship between the gray levels of the pixels immediately close in the image. Finally, the following features are extracted from both LE and RC sub-ROIs: contrast, correlation, cluster prominence, cluster shade, dissimilarity, energy, entropy, homogeneity, sum average, sum variance, sum entropy, difference entropy and normalized inverse difference moment.

#### 2.2.2. Principal Component Analysis

The feature extraction procedures returned five feature sets. The next step consisted of exploring the discriminating power of these sets to identify a sub-set of significant features. To do this, a feature reduction was performed through PCA [40].

The central idea of PCA is to reduce the dimensionality of a dataset consisting of a large number of interrelated variables while retaining as much as possible of the variation present in the data set. This is achieved by performing a linear transformation of the features that projects the original ones into a new Cartesian system, where the variables are sorted in descending order with respect to the overall variance percentage explained. In this work, a PCA was performed for each feature set.

The first step consists of representing the data set as a matrix. In particular, if n is the number of observations, in this case ROIs, and m the number of variables, a n × m matrix X is obtained. Then, the raw data in the matrix have to be standardized so that each variable contributes equally to the analysis. Thus, a new matrix Z is computed.

The next step consists of processing the correlation matrix R of Z, a m×m matrix whose elements are the correlation coefficients between the variables.

In general, the k-th principal component $y_{k}$ is such that:(1)$$y_{k}=Z\gamma_{k}\;\;\;\;\;\;\;\;\;\;\;\;\;\mathrm{and}\;\;\;\;\;\;\;\;\;var\left(y_{k}\right)=\lambda_{k}\;\;\;\;\forall k=2,\ldots ,m,$$
where $\lambda_{k}$ is the k-th largest eigenvalue of R and $\gamma_{k}$ is the corresponding eigenvector.

Now, a sub-set of principal components has to be selected to replace the m elements of Z by a much smaller number p of PCs. There are different rules for deciding how many PCs should be retained in order to account for most of the variation in Z, without serious information loss [40]. In this work, we adopted the explained variance criterion: the required number of PCs is the smallest value of p for which the chosen percentage, in this case 80%, is exceeded. In particular, since the principal components are sorted in descending order with respect to their variance, it is sufficient to select the p components whose summed variance exceeds the selected threshold value.

#### 2.2.3. Classification Model

Once the significant PCs were selected for each feature set, three different classification models were trained to discriminate the ROIs into benign and malignant, first on each PCs’ sub-set, then on the set containing all the selected PCs. We trained a random forest (RF) classifier [41], a naïve Bayes (NB) classifier [42] and a logistic regression (GLM) [43] in a procedure of PCs stepwise forward selection [44].

The RF algorithm is among the most popular machine learning algorithms because it generally provides good predictive performance combined with low over-fitting. In particular, the tree-based strategy used by RF naturally ranks by how well they improve the purity of the node: nodes with the greatest decrease in impurity are at the start of the trees, while nodes with the least decrease in impurity occur at the end of trees measured by Gini’s diversity index.

The NB algorithm is a probabilistic approach based on Bayes’ theorem with an assumption of independence among predictors. In simple terms, a naive Bayes classifier assumes that the presence of a particular feature in a class is unrelated to the presence of any other feature.

The GLM is a statistical model that uses a logistic function to model a binary variable. This regression transforms its output using the sigmoid function to return a probability value that can then be mapped to the two classes under question.

The stepwise forward selection algorithm identifies the best sub-set of PCs by sequentially adding a PC to the set selected in the previous steps. Starting from a model composed by a single PC that showed the highest median AUC [45] on 100 ten-fold cross-validation rounds, we iteratively added the PC that allowed us to obtain the highest classification performances in terms of median value AUC. The process was repeated until all variables were included into the model. The performance of each classification was evaluated on 100 ten-fold cross-validation rounds in order to obtain the variability of the experimental results.

Once the best model was selected for each method and dataset, we compared the classification performances of these prediction models also in terms of:*Accuracy* = (*TP* + *TN*)/(*TP* + *TN* + *FP* + *FN*)(2)
*Sensitivity* = *TP*/(*TP* + *FN*)(3)
*Specificity* = *TN*/(*TN* + *FP*)(4)
where *TP* and *TN* stand for true positive (number of true malignant ROIs identified) and true negative (number of true benign ROIs identified) cases, while *FP* (number of benign ROIs identified as malignant) and *FN* (number of malignant ROIs identified as benign) are the false positive and false negative ones, respectively. Specifically, the above values were calculated to identify the optimal threshold by means of Youden’s index on ROC curves [46], an index able to solve dataset unbalance problems (15 benign and 43 malignant).

## 3. Results

The aim of this study was to devise a prediction model that was successful in discriminating benign and malignant ROIs. First, we reduced the initial dataset dimensionality through a principal component analysis, and we obtained one sub-set of discriminant principal components for each initial feature set. Then, we used three different classification algorithms on both the individual obtained sub-set and on the complete sub-set of significant principal components.

### 3.1. Principal Component Analysis

We performed a PCA for each standardized set of features, then we adopted the explained variance criterion to select a sub-set of discriminant principal components. We chose a threshold value equal to 0.8.

The STAT set required 4 PCs, the GRAD set was represented by a set of 9 PCs, the COUNT set was replaced by a set of 3 PCs, the sub-set linked to the HAAR set was formed by a total of 19 PCs, and the GLCM set needed 11 PCs, as shown in Figure 5.

### 3.2. Classification Performances

Once the PCs’ subsets were extracted, we trained the random forest algorithm, the naïve Bayes method and the logistic regression in the procedure of PCs stepwise forward selection, first on each PCs’ sub-set, then on the set containing all the selected PCs.

Concerning the individual PCs’ sub-sets, the best models were obtained by training either the RF or the GLM classifier on sub-sets containing two PCs at most (Table 1).

Since the PCs were obtained as a linear combination of the starting variables, we were interested in understanding which of these features contributed most and positively to the computation of the PCs, which determined the best classification model for each set. We selected the variables shown in Table 2, excluding the ones characterized by coefficients close to zero compared to the those of the chosen variables.

As regards the STAT set, the best model had a median AUC of 83.49% and was obtained by training the GLM on the sub-set containing only the first PC, which was determined based on the contributions of the variables’ entropy, standard deviation, range, relative smoothness and variance, all computed on the RC images. The best model obtained with the GRAD set’s PCs reached a median AUC of 85.31% and included the RF classifier and the sub-set containing the first and the fifth PCs. These were estimated considering the variables’ mean, entropy, relative smoothness, variance, kurtosis and skewness computed on both the gradient’s magnitude and the gradient’s direction extracted from the RC and the LE images. Concerning the COUNT set, the GLM was the best classifier, achieving a median AUC of 75.66% when trained on the sub-set including the first and the third PCs. The most important features in the computation of these two PCs were the ones calculated on both the LE and the RC images by all five algorithms. As far as the HAAR set, the best model was obtained by training the RF classifier on the sub-set containing the second and the twelfth PCs. This model, with a median AUC of 94.65%, was the best, even compared to the best performing models of each set. The above-mentioned PCs were calculated mainly thanks to the contributions of the variables’ relative smoothness, entropy, skewness and kurtosis, which were all extracted from the LE images, except one extracted from the RC images, after the Haar decomposition. Finally, with regards to the GLCM set, a median AUC of 86.40% was reached with the RF classifier when trained on the sub-set including the first and the second PCs. The features that contributed most to their computation were entropy, sum entropy and sum average; the first were two extracted from the RC images while the third was extracted from the LE images.

As regards the models obtained by training the three classifiers on the complete set of PCs previously computed, the best was the one obtained with the RF classifier (Table 3).

Indeed, while the NB classifier reached a median AUC of 88.99% and the GLM achieved a median AUC of 90.08%, the model obtained by training the RF classifier on the set, including the second and the twelfth PCs belonging to the HAAR set and the first PC belonging to the GRAD set, reached a median AUC of 95.66% and a median accuracy of 90.52%. This model is also characterized by a better classification performance than the performance of the best model obtained by considering the single PCs’ sets, and reached a sensitivity of 88.37% and a specificity of 100%.

## 4. Discussion

Although its use is still not very widespread in the area, CESM is a very interesting technique in our opinion, and due to its intrinsic characteristics, it is particularly suitable for the analysis of images using radiomics. The opportunity to simultaneously analyze similar morphological images on the same mammographic and dynamic images with contrast medium for the evaluation of neoangiogenesis provides the system with a remarkable and varied series of information that can achieve excellent results. Moreover, the absence of a consistent number of studies on this method and on these issues, in our opinion, makes this work even more precious.

In this work, we proposed an automated support system able to characterize and discriminate breast lesions as benign/malignant. We extracted 58 ROIs from 53 CESM images and for each ROI we determined five set of features: the STAT set, which included statistical features extracted from the original ROIs, the GRAD and the HAAR set that comprised statistical measures extracted from the ROIs’ manipulations by filters and wavelet functions, respectively, the GLCM set that included textural features and the COUNT set, which comprised information about points and corners of interest. Subsequently, each above-mentioned feature set was replaced by the set containing the related discriminant principal components. In particular, for each set a principal component analysis was performed and a sub-set of principal components was selected by setting a variance explained threshold value of 0.8. Then, all the PCs’ sets obtained were used, first individually and then simultaneously, to train three different classifiers combined with a stepwise forward algorithm. The classification performances were evaluated and compared in terms of accuracy, sensitivity and specificity and AUC values on 100 ten-fold cross-validation rounds.

The best model among the ones developed on the single PCs’ sets turned out to be the model obtained by training the RF classifier on the HAAR PCs’ sub-set including the second and the twelfth PCs. It was observed that using only these two variables led to a median AUC value of 94.95%, a median accuracy of 87.93%, a sensitivity of 86.05% and a specificity of 100%.

On the other hand, it was observed that adding the first GRAD set’s PC to the previous model led to better results. Indeed, this new model reached a median AUC value of 95.66%, a median accuracy of 90.52%, a sensitivity of 88.37% and a specificity of 100%. In both cases, the specificity value highlights that these prediction models are able to correctly identify all the benign ROIs.

Thus, the sequential forward selection algorithm allowed us to obtain a very well performing classification model by selecting only three PCs: the second and the twelfth PCs from the HAAR set and the first PC from the GRAD set. The above-mentioned HAAR PCs were estimated thanks to the contributions of the variables, mainly calculated on LE images, only one of them was extracted from the RC images. On the other hand, the features that contributed most to the calculation of the first GRAD PC were all extracted from the LE images.

We compared the performance of the proposed approach with respect to the literature, and in this work, we improved the classification performances obtained with the CAD system developed in our previous work [21] (Table 4). Indeed, by using principal component analysis as a feature selection technique, instead of a backward feature selection algorithm combined with a naïve Bayes classifier, we reduced the number of discriminant variables and at the same time we obtained better results in terms of sensitivity and specificity (in the previous work we obtained a sensitivity of 87.5% and a specificity of 91.7%).

Moreover, compared to state of the art models, our model seems to perform better. In [19], the authors reached a sensitivity of 88% and a specificity of 92% by training a SVM classifier on a feature set extracted from 50 lesions manually segmented by radiologists.

Finally, we want to emphasize the importance of radiomic analysis for the characterization of benign and malignant ROIs and the achievement of more balanced values for sensitivity and specificity. Specifically, we want to compare our results with the ones obtained in [20] (a sensitivity of 100% and a specificity of 66%) with the use of only textural descriptors provided by the radiologist combined with CESM pixel information extracted directly from the images.

## 5. Conclusions

With the aim of improving the number of early diagnoses of breast cancer, in this work we proposed an automated expert system for discriminating benign and malignant ROIs. We proposed the use of a principal component analysis combined with machine learning techniques in order to select the optimal subset for characterizing breast regions and classifying them. Particularly, we trained three binary classifiers on an increasing number of features sorted by their diagnostic power, evaluated in terms of AUC. Our model’s performance represents a step forward compared to our previous work, both in regard to the greater accuracy achieved in the classification of benign and malignant lesions and the smaller number of variables used to obtain these results.

## Figures and Tables

**Figure 1 diagnostics-11-00684-f001:**
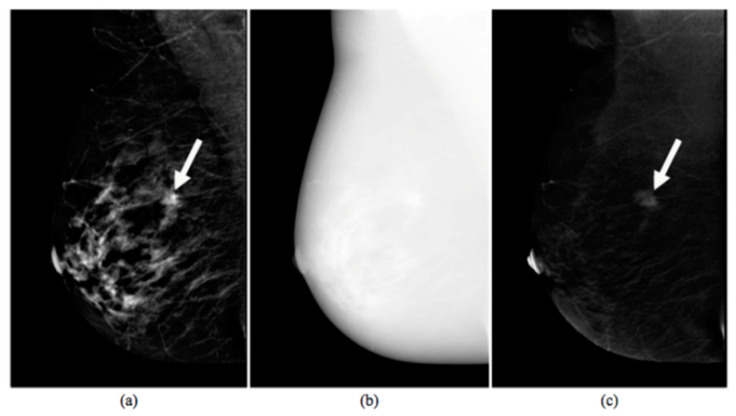
Images produced by contrast-enhanced spectral mammography (CESM instrumentation). Typical example of low energy (**a**), high energy (**b**), and recombined (**c**) images [17]. The white arrow points out a suspicious lesion.

**Figure 2 diagnostics-11-00684-f002:**
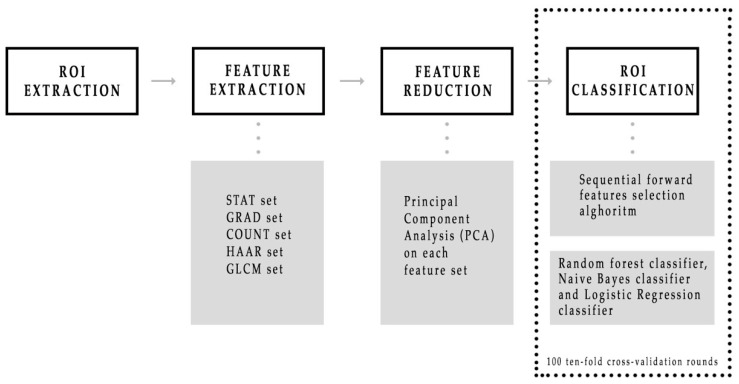
Schematic overview of the CESM images classification process. First, five sets of features were automatically extracted from each region of interest (ROI), then a principal component analysis was performed on each feature set. Finally, three binary classifiers were trained and their performances were evaluated on 100 ten-fold cross-validation rounds.

**Figure 3 diagnostics-11-00684-f003:**
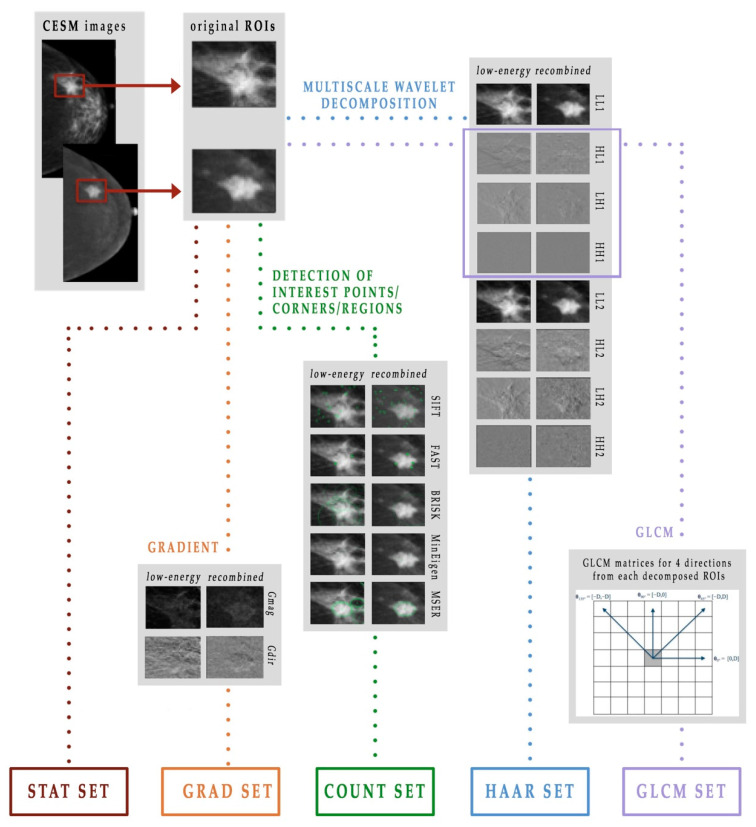
Scheme of the feature extraction process.

**Figure 4 diagnostics-11-00684-f004:**
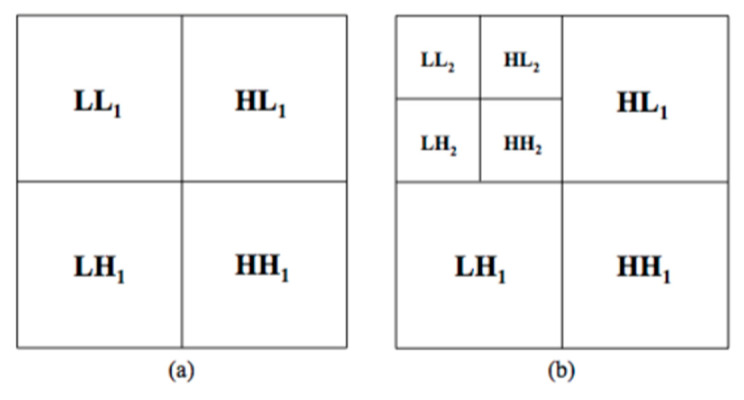
Haar decomposition: (**a**) first level and (**b**) second level decomposition.

**Figure 5 diagnostics-11-00684-f005:**
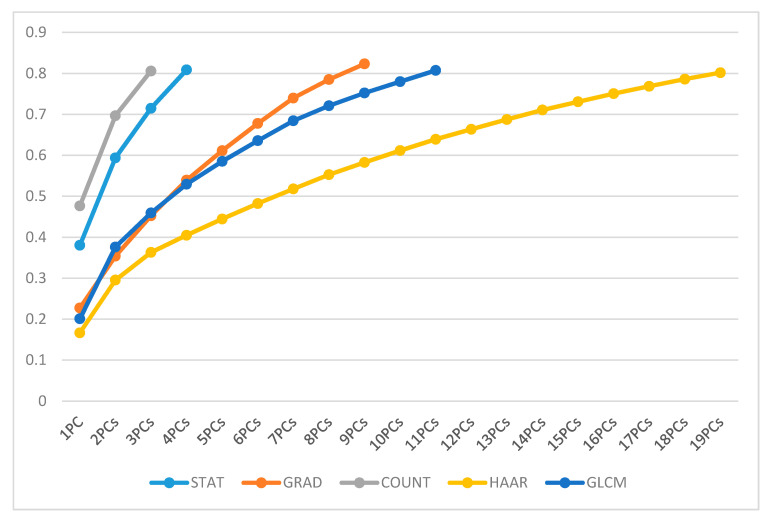
Overview of PC’s cumulative variance for each feature set.

**Table 1 diagnostics-11-00684-t001:** Classification performances of the best models obtained from the individual PCs’ sets calculated on 100 ten-fold cross-validation rounds. For each set of features, the performance measures of the different classification algorithms implemented are reported in correspondence with the best PC combination. with the best PC combination. The third column shows the PCs selected by the sequential forward selection algorithm. The best results are highlighted in bold.

PCs’ Set	Classifier	PCs Best Combination	AUC (%)	Acc (%)	Sens (%)	Spec (%)
STAT	RF	1 + 2	78.29	77.59	81.40	73.33
	NB	1	81.71	74.14	67.44	93.33
	**GLM**	**1**	**83.49**	**74.41**	**67.44**	**93.33**
GRAD	**RF**	**1 + 5**	**85.31**	**81.03**	**79.07**	**93.33**
	NB	1 + 2	76.59	75.86	76.74	73.33
	GLM	1 + 5 + 2 + 9	83.10	74.14	65.12	100
COUNT	RF	1 + 3	66.82	62.07	58.14	0.8
	NB	2 + 1	64.88	60.34	50.00	86.67
	**GLM**	**1 + 3**	**75.66**	**79.31**	**88.37**	**53.33**
HAAR	**RF**	**2 + 12**	**94.65**	**87.93**	**86.05**	**100**
	NB	1 + 3 + 16 + 19 + 15 + 14	86.51	84.48	87.21	80.00
	GLM	1 + 3 + 9 + 19 + 16 + 8 + 12	83.72	77.59	74.42	93.33
GLCM	**RF**	**2 + 1**	**86.40**	**81.03**	**79.07**	**86.67**
	NB	2 + 4 + 1 + 11 + 10 + 9	75.50	75.86	72.09	86.67
	GLM	2 + 4 + 1 + 9 + 11 + 10	82.33	87.93	93.02	73.33

**Table 2 diagnostics-11-00684-t002:** Overview of the features that were important in the computation of the selected PCs on 100 ten-fold cross-validation rounds. The factors, LE and RC identify the features extracted from the LE images and the RC images, respectively.

Set	PC	Important Features				
STAT	1	RC_Entropy	RC_Std	RC_Max-Min	RC_Relative Smoothness	RC_ Variance
GRAD	1	RC_Mean_ Gmag	LE_Mean_ Gmag	RC_Entropy_ Gmag	RC_Relative Smoothness_Gmag	RC_Variance_ Gmag
	5	LE_Kurtosis_ Gmag	LE_Skewness_ Gdir	RC_Skewness_ Gdir	RC_Kurtosis_ Gdir	RC_Entropy_ Gdir
COUNT	1	RC_Fast	RC_Brisk	LE_Brisk	LE_Fast	
	3	LE_Sift	RC_Sift	LE_MSER	RC_Minimum Eigenvalue	
HAAR	2	LE_Relative Smoothness_ HL2	LE_ Relative Smoothness_ LH2	LE_ Relative Smoothness_ HL1	LE_Entropy_ LL1	LE_Relative Smoothness_ HH2
	12	LE_Skewnes_ LL1	RC_Skewness_ LH2	LE_Kurtosis_ HL1	LE_Skewness_ HH1	
GLCM	2	RC_Sum Entropy_ HH1 dir2	RC_Entropy_ HH1 dir2	RC_Entropy_ HH1 dir3	RC_Entropy_ HH1 dir4	RC_Sum Entropy_ HH1 dir4
	1	LE_Sum Average_ HH1 dir3	LE_Sum Average_ HH1 dir1	LE_Sum Average_ HH1 dir4	LE_Sum Average_ HH1 dir2	

**Table 3 diagnostics-11-00684-t003:** Classification performance of the best models obtained from the complete set of PCs calculated on 100 ten-fold cross-validation rounds. The best result is highlighted in bold.

Classifier	Best Model	AUC (%)	Acc (%)	Sens (%)	Spec (%)
**RF**	**H2 + G1 + H12**	**95.66**	**90.52**	**88.37**	**100**
NB	S1 + GL10 + G2 + GL11 + H16 + H3 + H19	88.99	89.66	93.02	80
GLM	S1 + G2 + G9 + S3 + H8 + GL2 + GL1	90.08	84.48	81.40	100

**Table 4 diagnostics-11-00684-t004:** Benign vs. malignant breast lesion classification evaluated on CESM images: comparison of the performance results of the proposed models in the literature.

Article	No. of ROIs	Features	Classifier	Performance (%)
Patel et al. [19]	50		SVM	AUC: 95 Acc: 90 Sens: 88 Spec: 92
Perek et al. [20]	129		Multimodal Network	AUC: 89 Sens: 100 Spec: 66
Fanizzi et al. [21]	48	12	Random Forest	AUC: 93.1 Acc: 87.5 Sens: 87.5 Spec: 91.7
Losurdo et al. [23]	55	10	SVM	Acc: 80.91 Sens: 90.28 Spec: 71.55
Best proposed model	58	2	Random Forest	AUC: 95.66 Acc: 90.52 Sens: 88.37 Spec: 100

## Data Availability

The data presented in this study are available on request from the corresponding author. The data are not publicly available because they are the property of Istituto Tumori ‘Giovanni Paolo II’—Bari, Italy.

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
