# Peer review of "Radiomic Feature Reduction Approach to Predict Breast Cancer by Contrast-Enhanced Spectral Mammography Images"

_diagnostics, 2021, doi:10.3390/diagnostics11040684_

Round 1

Reviewer 1 Report

In this work, authors describe  a large dataset of radiomic on which the feature selection techniques can influence the process and accuracy of cancer classification. Abstract is well organized. The Introduction is sufficient. material and methods are well described. Results are representative. Figures and Tables are good quality. I found this paper show very important and valid data and my recommendation is to accept.

Author Response

Point 1. In this work, authors describe a large dataset of radiomic on which the feature selection techniques can influence the process and accuracy of cancer classification. Abstract is well organized. The Introduction is sufficient. material and methods are well described. Results are representative. Figures and Tables are good quality. I found this paper show very important and valid data and my recommendation is to accept.

Response 1.We have revised the introduction. We thank the reviewer.

Reviewer 2 Report

The paper suggests a traditional machine learning to discriminate benign and malignant breast lesions.

  • The authors states that: “ the literature is poor in CAD systems for breast lesions using CESM images”. A quick review on Google Scholar using as key words: cad + breast lesion + cesm images gave back around 85 papers. Only 3 papers are cited, with two from the own authors
  • It is not clear what is the difference between the current work and the previous ones [31,33]
  • What is the difference between “a sub-set of these features, characterized by a higher discriminating power, is selected for a more manageable data processing” and the next step “selected the most important features”
  • Please provide further information related with the manual lesion delineation. It was performed by using any assistant tools? Using 2D or 3D? How many lesions per subject?
  • Why the authors performed: 100 ten-fold cross validation rounds
  • Any pre-processing was performed in the images?
  • For the GLCM computation; what was the the chosen D parameter?
  • About the feature selection, PCA, how many feature were selected per feature set?
  • The introduction must be review to include a more comprehensive literature review
  • The methodology must be review to include further detail about the experiments, parameters selection.
  • In the results, please include related results in the literature for comparison (even the previous papers). Benchmarking is necessary 

Author Response

Point 1. The paper suggests a traditional machine learning to discriminate benign and malignant breast lesions. The authors states that: “the literature is poor in CAD systems for breast lesions using CESM images”. A quick review on Google Scholar using as key words: cad + breast lesion + cesm images gave back around 85 papers. Only 3 papers are cited, with two from the own authors

Respose 1. We thank the reviewer for the work done. We understand the reviewer's observation. However, of the articles identified by the research carried out by the reviewer, only 2 are explicitly concerned the theme of our work (already present in our manuscript, i.e. ref 27-28), that is the development of expert systems as support diagnostic tool for CESM images. Indeed, the most of recent literature works are comparative studies on the diagnostic performance of human readers on CESM images with respect to those of mammographic and MR images. On the contrary, the state-of-the-art is poor about the development of expert systems. For this reason, our interest for this trend and in particular for this innovative instrumentation is strong. In the manuscript we have cited the papers have developed the support tools for CESM images, such as Petal et al [27] and Peker et al [28] other that our previous preliminary work [31, 33] and compared our experimental results with they.

Point 2. It is not clear what is the difference between the current work and the previous ones [31,33]

Respose 2. In our previous works, in order to reduce the dataset dimensionality we have implemented several feature selection techniques, such as filtered, embedded and wrapped methods, or a combination of them. However, although the subset of features identified was the one with the greatest discriminating power with respect to the various evaluation criteria considered, the feature selection techniques involve a natural and inevitable loss of information due to the exclusion of features from the original dataset. For this reason, before proceeding with the exploration of new features useful for increasing the accuracy of the classification performance, we considered it appropriate to evaluate a different dimensionality reduction technique that does not produce information losses but only a reduction of the noise inherent in the dataset. We then carried out an analysis of the main components and developed a prediction model based on a subset of them appropriately selected to achieve optimal performance. We specified this in the introduction of the manuscript and compared the results obtained with the previous work.

Point 3. What is the difference between “a sub-set of these features, characterized by a higher discriminating power, is selected for a more manageable data processing” and the next step “selected the most important features”

Response 3.  In previous works, we have applied a combination of different feature selection techniques. In the specific, we first made a first selection by identifying the statistically significant ones by means of non-parametric statistical tests (filtering methods). Afterwards, on this subset were multivariate feature selection techniques. We have better explained this difference in the manuscript.

Point 4. Please provide further information related with the manual lesion delineation. It was performed by using any assistant tools? Using 2D or 3D? How many lesions per subject?

Response 4. The radiologist indicated the ROIs on both LE and Recombined (RC) by means of a box on the image using the reporting tool. 5 patients had two lesions. We have emphasized this in the manuscript.

Point 5. Why the authors performed: 100 ten-fold cross validation rounds.

Response 5.We performed 100 ten-fold cross validation rounds in order to have a statistic on the variability of the experimental results obtained. We have emphasized this in the manuscript.

Point 6. Any pre-processing was performed in the images?

Response 6.On the ROI the features (HAAR set) on the pre-processed image by Haar transform were calculated as described in sub-paragraph 2.2.1. No other pre-processing was performed.

Point 7. For the GLCM computation; what was the the chosen D parameter?

Response 7.Sorry, it is a typo. We have corrected.

Point 8. About the feature selection, PCA, how many feature were selected per feature set?

Response 8. Table 1 shows for each set of features, the performance measures of the different classification algorithms implemented in correspondence with the best PC combination. The third column shows the PCs selected by the sequential forward selection algorithm. We have better described the contents of the table in its caption.

Point 9. The introduction must be review to include a more comprehensive literature review.

Response 9. We revised the introduction according to the auditor's instructions. We have included some new articles recently proposed about CESM imaging support systems but aimed at predicting the response to neoadjuvant therapy. On the other hand, compared to what we have previously identified, we have not found new articles relating specifically to diagnostic models.

Point 10. The methodology must be review to include further detail about the experiments, parameters selection.

Response 10. In the subparagraph 2.2.3, we have better described the stepwise foreword procedure for selecting the optimal PCs sub-set.

Point 11. In the results, please include related results in the literature for comparison (even the previous papers). Benchmarking is necessary 

Response 11. We have inserted a table for comparing the classification results of the model at the state of the art.

Round 2

Reviewer 2 Report

The authors answered all the questions/ concerns that I pointed out in my previous review. 

My only suggestion is to include the parameter D used for the co-occurrence matrix. It was not a typo, it is a parameter when computed the matrix, the distance between the pixels.

Author Response

Point 1. My only suggestion is to include the parameter D used for the co-occurrence matrix. It was not a typo, it is a parameter when computed the matrix, the distance between the pixels.

Response 1.We thank the reviewer for the advice. In the previous version D was actually a typo and we did not understand he asked us to specify the offset value. However, for completeness, we have included again the parameter D and we have specified its value.